# An Analytic Solution to Precipitation Attenuation Expression with Spaceborne Synthetic Aperture Radar Based on Volterra Integral Equation

Ting Luo, Yanan Xie, Rui Wang * and Xueying Yu

School of Communication and Information Engineering, Shanghai University, Shanghai 200444, China;
alisa_luo@shu.edu.cn (T.L.); yxie@shu.edu.cn (Y.X.); weareone17@shu.edu.cn (X.Y.)
* Correspondence: rwang@shu.edu.cn; Tel.: +86-158-0084-0930

**Abstract:** Precipitation is closely related to the production and daily life of human beings, so accurate precipitation measurement is of great significance. Spaceborne synthetic aperture radar (SAR) is a microwave remote sensing technology with high resolution, which provides an opportunity to improve the accuracy of precipitation inversion. In this paper, the radar attenuation expression is analyzed according to the scattering characteristics of rain, snow and ground. Combined with the Volterra integral equation of the second kind, the solution to the expression, the precipitation horizontal variation of the double-layer model, can be obtained. The simulated result of this method is in good agreement with the given horizontal variation of precipitation. Compared with the original VIE method, which only considers the effect of rainfall, the method in this paper considers both rainfall and snowfall; compared with the Model Oriented Statistical (MOS) method, the method in this paper not only reduces the number of empirical coefficients used and thus reduces the workload in the early stage and retrieval process and its application limits, but it will also increase the accuracy of the inversion of the horizontal variation.

**Keywords:** precipitation measurement; synthetic aperture radar; Volterra integral equation; attenuation expression

## 1. Introduction

Precipitation is a phenomenon that the water vapor in the atmosphere condenses and falls to the ground in the form of liquid water or solid water. It includes rain, snow, sleet, frost, hail, ice particles, etc. Human production and activities correlate with precipitation; thus, the accurate measurement of precipitation is particularly necessary [1,2]. At present, however, only a small number of countries can obtain relatively accurate distribution information of the global precipitation. Therefore, an increasing number of studies need to be carried out on global precipitation measurement.

Remote sensing is a comprehensive earth observation technology developed in the 1960s. Remote sensing technology has made great progress, and its application has been gradually more extensive since the 1980s. According to the principle that the ability to absorb and reflect electromagnetic waves differ from object to object, once such abilities of an object on the surface are detected, information can be extracted to identify the object from a long distance away, which offers potential to precipitation observations.

Synthetic aperture radar (SAR) is an all-weather, all-time high-resolution microwave imaging radar. It applies a synthetic aperture principle, pulse compression technology and a signal processing method, aiming at obtaining azimuth high-resolution and range high-resolution imaging with a real aperture antenna under a wide frequency range and different polarizations. The SAR system, providing useful information to human beings, has been extensively used in military, economic, scientific and technological fields, and has broader application prospects and greater development potential [3].

The frequency of radio waves working at X band is only 4 GHz lower than that working at Ku band, the working frequency band of precipitation radar (PR). Furthermore, studies have shown that X-SARs are more sensitive to precipitation than SARs operating under lower frequencies [4,5]. Many countries have successfully launched SAR systems carrying high-frequency X-band synthetic aperture radar (X-SAR) sensors, such as SRTM launched by America in 2000, Cosmo-Skymed launched by Italy in 2006, TerraSAR-X launched by German in 2007, etc. The benefits of using X-SAR to retrieve precipitation are as follows. First, the scale of the spaceborne radar antennas is limited, leading to a serious non-uniform beam filling effect and bringing great deviation to precipitation inversion [6]. Fortunately, this problem can be solved with a high-resolution X-SAR and microwave radiation measurement data. Second, precipitation measurement in mountainous areas is also an important topic. The terrain of such areas brings challenges to PRs while X-SARs provides a method for precipitation measurement in mountainous areas. Third, X-SARs can observe mid-latitude storms which Tropical Rainfall Measuring Mission (TRMM) might miss [7]. Thus, the observation results of X-SARs will supplement and enhance those of Global Precipitation Mission (GPM). Fourth, X-SARs used in the precipitation retrieval increases practical use of X-SAR satellite images and can improve the investment benefits of X-SARs.

Pichugin et al. established a single-layer, the rainfall layer, model for radar observation in 1991 [8]. They obtained the analytical solution to the horizontal rainfall variation of a single-layer model by solving the rainfall attenuation expression combining the Volterra integral equation (VIE) of the second kind. However, because the attenuation coefficients of different condensates (such as rain and snow) are different, this method cannot be directly applied to the model containing two or more condensates. Marzano et al. proposed a Model Oriented Statistical (MOS) method. They considered a more complex double-layer precipitation model with a snowfall layer and a rainfall layer completely separated in 2008 [9,10]. They analyzed the characteristics of radar backscattering cross section and obtained the statistical solution to the horizontal precipitation variation of a double-layer rainfall model. However, this method relies on many empirical values, thus having a large workload in the early stage. What makes the result less accurate is that it makes an approximation in retrieval of the shape of the precipitation area. Xie et al. proposed a MOSVI method, combining the MOS method and the VIE method [11]. They apply the VIE method to obtain information on the shape of the precipitation area and the MOS method to calculate the rain rate. It partially reduces the numbers of the empirical values of MOS method. To further reduce the numbers of the empirical values so as to reduce the workload in the early stage, an analytical method to obtain the horizontal variation of the double-layer precipitation model is proposed in this paper.

## 2. Principles and Model Establishment

The radar antenna only receives the energy that returns to the radar, which is called the backscattering energy. The total scattering power of a scattering particle is obtained by multiplying the incident wave energy flow density by the radar cross section (RCS). When the scattering particle makes isotropic scattering with this total power, the power density scattered to the antenna is just equal to the actual backscattering energy flow density caused by the particle at the antenna [12]. Thus, we can obtain the backscattering energy of the particle from its normalized radar cross section (NRCS) which can be divided into the surface backscattering cross section and the precipitation backscattering cross section:

$$\sigma_{SAR} = \sigma_{srf} + \sigma_{vol}, \tag{1}$$

where $\sigma_{SAR}$ is the radar backscattering cross section, $\sigma_{srf}$ is the surface backscattering cross section, $\sigma_{vol}$ is the volume backscattering cross section.

### 2.1. The Precipitation Cross Sectional Model

To simplify the precipitation model and reduce the method complexity, a lot of assumptions are made as follows. The microwave pulses emitted by the radar are treated as plane waves. Among all the hydrometeors, only the rainfall and snowfall are considered and seen as independent, which means the melting layer is ignored. The ground is considered as a plane terrain. The simplified cross sectional model of rainfall and snowfall is shown in Figure 1.

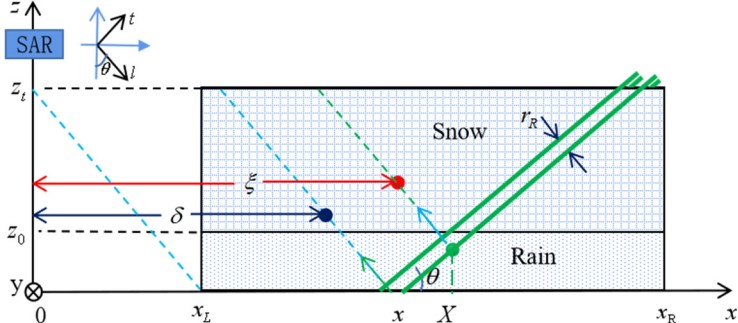

**Figure 1.** Simplified cross sectional model of precipitation. The $x$-axis is the cross-track direction, the $y$-axis is the along-track direction, the $z$-axis is the altitude direction, the $l$-direction is the propagation direction of microwaves emitted by the radar, and the $t$-direction is the direction of the reflected microwaves, perpendicular to the $l$-direction. $z_t$ is the height of the rain-cloud cell in km and $z_0$ is the boundary height of the rainfall and snowfall in km. $\theta$ is the slant off-nadir angle. The microwave pulses emitted by the radar are considered as plane wave front slices, shown as a pair of lines with a width of $r_R$. $\delta$ is the $x$-coordinate of the point on the path of the backscattering echo caused by ground. $\xi$ is the $x$-coordinate of the point on the path of the backscattering echo caused by hydrometeors. $X$ is the $x$-coordinate of the point on the microwave front slices. Rectangular rain-cloud cell is chosen just for simplicity and more shapes will be used in the simulation section.

### 2.2. The Precipitation Spatial Distribution

The precipitation spatial precipitation rate is simplified as the factorization of the vertical distribution variation and its horizontal variation [9]:

$$R(x,z) = I(x) \cdot V(z), \tag{2}$$

where $R(x,z)$ is the precipitation rate in mm/h, $V(z)$ is the vertical distribution of the precipitation and $I(x)$ is the horizontal variation of $V(z)$. The along-track resolution $\cdot y$ is estimated to be 264 m [10]. This degraded along-track resolution is adequate for hydrological purposes. This paper will focus on the solution to the horizontal variation $I(x)$. Thus, the vertical distribution $V(z)$ will be considered uniform for simplicity:

$$V(z) = V(0), \ 0 \leq z \leq z_t, \tag{3}$$

where $V(0)$ is the near-surface precipitation rate in mm/h. $I(x)$ representing a single rain-cloud cell can be expressed as:

$$I(x) = \begin{cases} 0, & 0 \leq x < z_t/\tan\theta. \\ x/d - z_t/(d\tan\theta), & z_t/\tan\theta \leq x < d + z_t/\tan\theta. \\ 1, & d + z_t/\tan\theta \leq x < w - d + z_t/\tan\theta. \\ (w-x)/d + z_t/(d\tan\theta), & w - d + z_t/\tan\theta \leq x < w + z_t/\tan\theta. \\ 0, & x \geq w + z_t/\tan\theta. \end{cases} \tag{4}$$

where $d$ is the shape parameter of the horizontal variation of the precipitation, $w$ is the width of the precipitation area in km.

In this paper, three common cases for the horizontal variation with a single rain-cloud cell are used. For $d = 0$:

$$I(x) = \begin{cases} 0, & 0 \leq x < z_t / \tan\theta. \\ 1, & z_t / \tan\theta \leq x < w + z_t / \tan\theta., \\ 0, & x \geq w + z_t / \tan\theta. \end{cases} \tag{5}$$

representing the rectangular variation.

For $d = w/2$:

$$I(x) = \begin{cases} 0, & 0 \leq x < z_t / \tan\theta. \\ (x - z_t tan\theta)/(w/2), & z_t / \tan\theta \leq x < w/2 + z_t / \tan\theta. \\ (w - x + z_t tan\theta)/(w/2), & w/2 + z_t / \tan\theta \leq x < w + z_t / \tan\theta. \\ 0, & x \geq w + z_t / \tan\theta. \end{cases} \tag{6}$$

representing the triangular variation.

For $0 < d < w/2$, $I(x)$ represents the trapezoidal variation.

To specify, $I(x)$ representing two or more rain-cloud cells can be realized by superimposing two or more single rain-cloud cells accordingly. In this paper, the rectangular variation with two separate rain-cloud cells is also considered in the simulation and it can be expressed as:

$$I(x) = \begin{cases} 0, & 0 \leq x < z_t / \tan\theta \\ 1, & z_t / \tan\theta \leq x < d + z_t / \tan\theta \\ 0, & d + z_t / \tan\theta \leq x < w - d + z_t / \tan\theta \\ 1, & w - d + z_t / \tan\theta \leq x < w + z_t / \tan\theta \\ 0, & x \geq w + z_t / \tan\theta \end{cases} \tag{7}$$

### 2.3. Normalized Radar Cross Section Analysis

Assume that the average surface backscattering cross section of ground is $\sigma_0$. When the radar beam enters the area with hydrometeors which are rain and snow in this paper, the surface scattering echoes will be attenuated:

$$\sigma_{srf}(x) = \sigma_0 \cdot L^2(x), \tag{8}$$

where $L$ is the one-way atmospheric loss factor, relevant to the path of the radar beam. Then, Equation (8) can be rewritten as:

$$\sigma_{srf}(x) = \sigma_0 \cdot exp(-2/\sin\theta \int_{x_{s_1}}^{x_{s_2}} k(\delta)d\delta), \tag{9}$$

where $x_{s_1}$ and $x_{s_2}$ are the $x$-coordinates of the left and right limits of the attenuation path of the scattering echo along $l$-direction caused by ground.

Similarly, when the radar beam enters the area with hydrometeors, the precipitation scattering echoes will be attenuated:

$$\sigma_{vol}(x) = \tan\theta \int_{x_{v_1}}^{x_{v_2}} L^2(x) \cdot \eta(X)dX, \tag{10}$$

where $x_{v_1}$ and $x_{v_2}$ are the $x$-coordinates of the left and right limits of the attenuation path of reflecting along $t$-direction caused by meteorological particles. Furthermore, it can also be rewritten as:

$$\sigma_{vol}(x) = \tan\theta \int_{x_{v_1}}^{x_{v_2}} exp\left(-2/\sin\theta \int_{x_{v_3}}^{x_{v_4}} k(\xi)d\xi\right)\eta(X)dX, \tag{11}$$

where $x_{v_3}$ and $x_{v_4}$ are the $x$-coordinates of the left and right limits of the attenuation path of scattering echo along $l$-direction caused by meteorological particles. $\eta$ is the radar reflectivity, defined as the summation of radar cross sections of all precipitation particles per unit volume, relevant to the distribution and size of the meteorological particles and the frequency of the microwaves. A widely accepted way to express $\eta$ [13,14] is that:

$$\eta(x) = \frac{\pi^5 |K_0|^2}{\lambda^4} i_1 [R(x)]^{b_1} = a_1 [R(x)]^{b_1},\qquad(12)$$

where $|K_0|^2$ is an expression of negative refractive index, which is approximately 0.19 for snow and 0.93 for rain when neglecting the influence of the temperature. Furthermore, $\lambda$ is the wavelength of electromagnetic waves, which is about 3.1 cm for X-band SAR. $i_1$ and $b_1$ are empirical values. Lots of studies have been carried out on the values of $i_1$ and $b_1$. In this paper, the values calculated by Ulaby [15] (pp. 318–328) are adopted. For rain, $i_1$ is 300, $b_1$ is 1.1; for snow, $i_1$ is 182, $b_1$ is 1.4.

### 2.4. Extinction Coefficient k

Scatterers (rain, snow and other hydrometeors) are generally considered to be randomly distributed, and assume that there is no coherent phase relationship between the scattering fields of each single particle. Therefore, the volume absorption and scattering containing many particles can be calculated through incoherent scattering theory. According to the different attenuation effects of rainfall and snowfall, the attenuation coefficient $k$ will be expressed by $k_r$ and $k_s$ respectively. For a given droplet size distribution $p(r)$, where $r$ is the droplet size, the rainfall extinction coefficient can be calculated [15] (pp. 318–326):

$$k_r(x) = \frac{\lambda^3}{8\pi^2} \int_0^\infty \chi^2 p(\chi) \varsigma_e(\chi) d\chi,\qquad(13)$$

$$\chi = \frac{2\pi r}{\lambda},\qquad(14)$$

where $\varsigma_e$ is the attenuation coefficient of Mie Scattering. Furthermore, the snowfall extinction coefficient can be calculated [15] (pp. 326–328):

$$k_s(x) = 4.34 \times 10^{-4} \left[ \frac{2 \times 10^{-3} \pi^5}{3\lambda^4 \rho_s^2} |K_{ds}|^2 \sum_{i=1}^{N_v} d_i^6 + \frac{\pi^2}{\lambda \rho_s} Im\{-K_{ds}\} \sum_{i=1}^{N_v} d_i^3 \right],\qquad(15)$$

where $\rho_s$ is the density of snowfall, $K_{ds}$ is related to the refraction of dry snow, $d_i$ is diameter of snowfall, $N_v$ is the total number of droplets per unit volume.

Such theoretical method of calculating the extinction coefficient of rainfall and snowfall are widely proved effective and many scholars have calculated the values under a wide range of frequencies [16–18]. However, due to its complexity, this method is difficult to be directly applied to a certain algorithm. To simplify the acquisition of the extinction coefficients, Ryde [19] proposed a linear relationship between the extinction coefficients and the precipitation:

$$k(x) = a \cdot R(x),\qquad(16)$$

where $R$ is the precipitation in mm/h, $a$ is an empirical value. Later, a logarithmic relationship between the extinction coefficients and the precipitation is proposed and used [20–22]:

$$k(x) = a \cdot R(x)^b,\qquad(17)$$

where $a$ and $b$ are both empirical coefficients. Scholars have conducted extensive research on the empirical coefficients of Equations (16) and (17) [23–28]. Furthermore, their root mean square (rms) errors are listed in Table 1. It is obvious that the rms errors of the linear model are slightly larger than those of the logarithmic model. However, using the logarithmic model will largely increase the complexity of the method introduced in this

paper and the rms error of the linear model is acceptable, so in this paper, the linear model is adopted. The values calculated by Ulaby [15,21] are adopted. For rain, $a = 3.349 \times 10^{-3}$; for snow, $a = 2.229 \times 10^{-3}$.

**Table 1.** The root mean square(rms) error of the linear model in Equation (16) and the logarithmic model in (17) [15].

| Frequency (GHz) | rms Error of (17) (Percent) | rms Error of (16) (Percent) |
|:---:|:---:|:---:|
| 7.5 | 28 | 31 |
| 9.4 | 30 | 36 |
| 16.0 | 22 | 28 |
| 34.9 | 10 | 12 |

*2.5. Analysis of Radar Backscattering Echo Path*

Based on the cross sectional model established in Section 2.1, it is convenient to analyze the radar backscattering echo path, as is shown in Appendix A. Furthermore, explicit NRCS formulas under different precipitation ranges can be derived with the analysis of the paths of the radar backscattering echoes.

According to the analysis of the path of the radar echoes and the effect of precipitation on radar echoes, it is obvious that the paths of different radar echoes, the extinction coefficients and the radar reflectivity are different under different precipitation ranges. Thus, attention should be paid during the simulation.

## 3. Volterra Integral Equation (VIE) of the Second Kind Inversion Method

Since the extinction coefficient $k$ and the radar reflectivity $\eta$ will be zero, according to Equations (12) and (16), if the precipitation rate is zero, we can easily integrate the numerous formulas in Appendix A into one:

$$
\begin{aligned}
\sigma_{SAR}(x) = {} & \sigma_0 exp\left[-\frac{2}{\sin\theta}\left(\int_{x-z_t\tan\theta}^{x-z_0\tan\theta} k_s(\delta)d\delta + \int_{x-z_0\tan\theta}^{x} k_r(\delta)d\delta\right)\right] + \\
& \tan\theta \int_{x+z_0/\tan\theta}^{x+z_t/\tan\theta} exp\left(-\frac{2}{\sin\theta}\int_{X-[z_t-(X-x)\tan\theta]\tan\theta}^{X} k_s(\xi)d\xi\right)\eta_s(X)dX + \\
& \tan\theta \int_{x}^{x+z_0/\tan\theta} exp\left[-\frac{2}{\sin\theta}\left(\int_{X-[z_t-(X-x)\tan\theta]\tan\theta}^{X-[z_0-(X-x)\tan\theta]\tan\theta} k_s(\xi)d\xi + \int_{X-[z_0-(X-x)\tan\theta]\tan\theta}^{X} k_r(\xi)d\xi\right)\right]\eta_r(X)dX.
\end{aligned}
\tag{18}
$$

This nonlinear formula, however, is not easy to deal with. To simplify this formula, we can introduce a function:

$$
p(x) = exp\left(\frac{2}{\sin\theta}\int_0^x k_r(m)dm\right). \tag{19}
$$

Using $p(x)$, the extinction coefficients of rain, the radar reflectivity of rain and snow, together with the precipitation rate can also be expressed, as is derived in Appendix B. After the substitution and transformation, also shown in Appendix B, we can obtain:

$$
\begin{aligned}
p(x)^{\frac{a_s}{a_r}} = {} & f_1(x) - \lambda_1 \int_{x+z_0/\tan\theta+z_0\tan\theta}^{x+z_t/\tan\theta+z_t\tan\theta} K_3(X)p(t)^{\frac{a_s}{a_r}}dt \\
& -\lambda_2 \int_x^{x+z_0/\tan\theta+z_0\tan\theta} K_4(X)p(t)^{\frac{a_s}{a_r}}dt,
\end{aligned}
\tag{20}
$$

where

$$
f_1(x) = \frac{\sigma_{SAR}(x+z_t\tan\theta)\cdot p(x+z_t\tan\theta)\cdot p[x+(z_t-z_0)\tan\theta]^{\frac{a_s}{a_r}-1}}{\sigma_0}, \tag{21}
$$

$$
K_3(X) = K_1(X)\cdot p[x+(z_t-z_0)\tan\theta]^{\frac{a_s}{a_r}-1}\cdot p(x+z_t\tan\theta), \tag{22}
$$

$$
K_4(X) = K_2(X)\cdot p[x+(z_t-z_0)\tan\theta]^{\frac{a_s}{a_r}-1}\cdot p(x+z_t\tan\theta), \tag{23}
$$

$$
\lambda_1 = \frac{G_1\cos^2\theta}{\sigma_0}, \tag{24}
$$

$$\lambda_2 = \frac{G_2 \cos^2 \theta}{\sigma_0}. \tag{25}$$

Assume that for $x \geq x_m$, there is no precipitation; that is $I(x) = 0$. According to Equations (16) and (19), for $I(x) = 0$, we can always obtain $k(x) = 0$ and $p(x) = 1$. Then, we can obtain the values of $K_3(X)$, $K_4(X)$, $K_1(X)$, $K_2(X)$ and $f_1(x)$ for $x_m - (z_t - z_0) \tan \theta \leq x \leq x_m$. Furthermore, naturally, we can obtain $p(x)$ for $x_m - (z_t - z_0) \tan \theta \leq x \leq x_m$. Furthermore, we can then obtain $p(x)$ for $x_m - (n+1)(z_t - z_0) \tan \theta \leq x \leq x_m - n(z_t - z_0) \tan \theta$, where n is a non-zero natural number. Through such iterations, we can obtain $p(x)$ under ranges of interest. The iteration processes are explicitly shown in Appendix B.

## 4. Simulation Results

For X-band SAR working at the oblique angle of $30°$, $\sigma_0$ is approximately $-7$ dB [29]. Set $z_t = 13$ km and $z_0 = 4.5$ km. From Equations (12), (16) and (18), we can first obtain the normalized radar cross section (NRCS) $\sigma_{SAR}$ of the double-layer precipitation model. With Formulas (6) or (7), we can obtain the horizontal precipitation variation of the model. Taking a rectangular rain-cloud cell, a trapezoidal rain-cloud cell, a triangular rain-cloud cell and two separate rectangular rain-cloud cells under the rain rate of 10 mm/h as Examples 1–4, the precipitation variation simulation result diagrams, given precipitation variation diagrams, comparison diagrams and absolute error diagrams are shown in Figures 2–5. Taking a triangular rain-cloud cell under the rain rate of 30 mm/h and 50 mm/h as Examples 5–6, the precipitation variation simulation result diagrams, given precipitation variation diagrams, comparison diagrams and relative error diagrams are shown in Figures 6 and 7. The attributes of the six examples, including the maximum rain rate $R(x)$, the width of the rain-cloud cells $w$, the shape parameters $d$ and the shapes of rain-clouds are listed in Table 2. The simulation results of the attributes are listed in Table 3, where $\hat{R}(x)$ is the simulated result of the maximum rain rate, $\widetilde{R}(x)$ is the relative error of the maximum rain rate, $\hat{w}$ is the simulated result of the width of the rain cloud and $\widetilde{w}$ is the relative error of the width of the rain cloud.

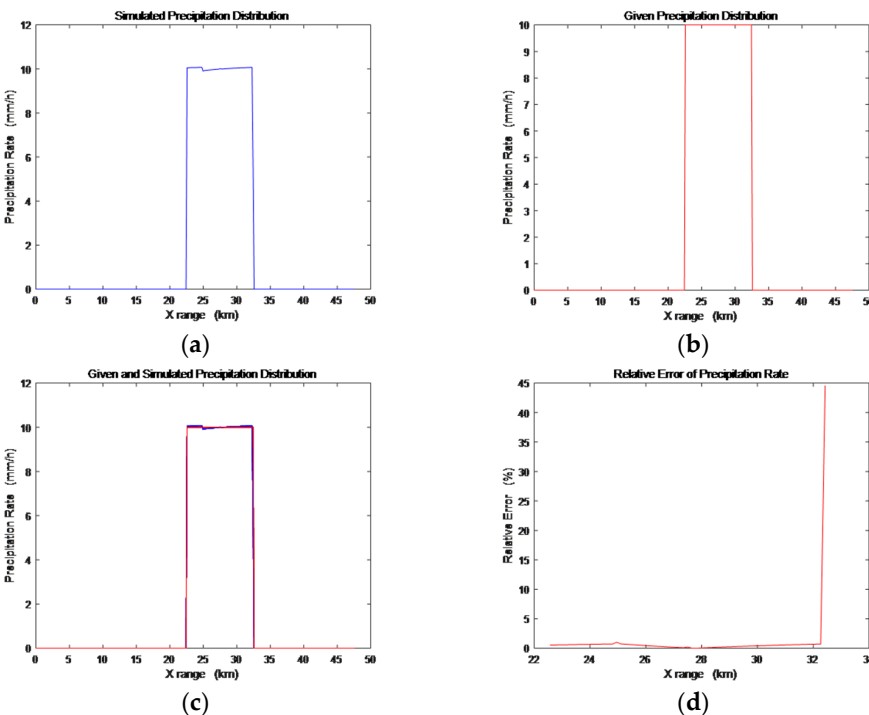

**Figure 2.** Rectangular distribution diagrams for $R(x) = 10$ mm/h. (**a**) Given precipitation distribution; (**b**) Simulated precipitation distribution; (**c**) Given precipitation distribution and simulated precipitation distribution; (**d**) Relative error between given and simulated result.

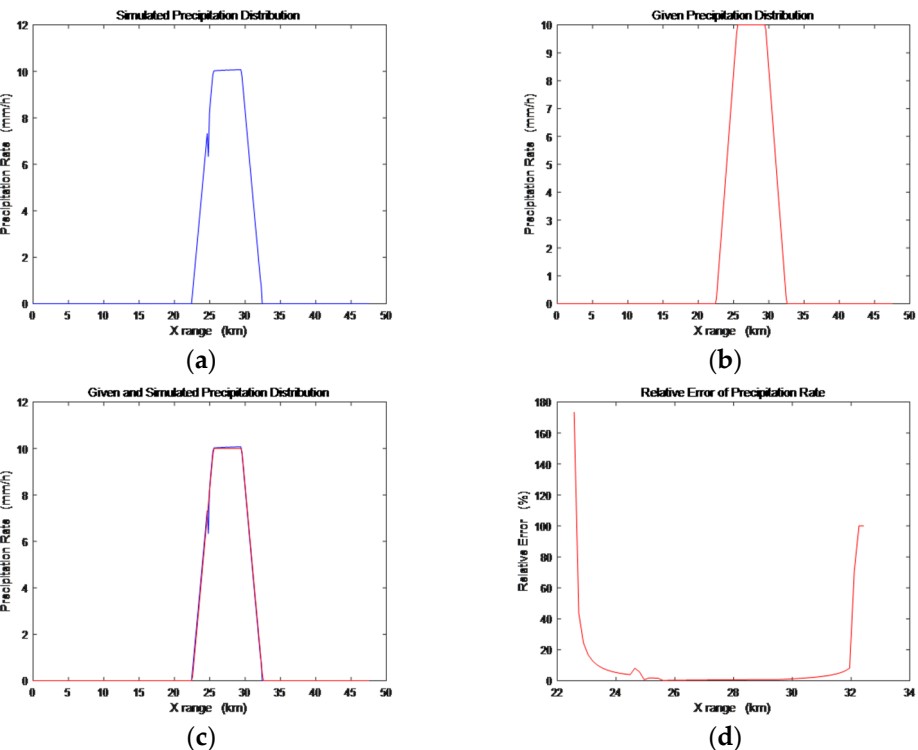

**Figure 3.** Trapezoidal distribution diagrams for $R(x) = 10$ mm/h. (**a**) Given precipitation distribution; (**b**) Simulated precipitation distribution; (**c**) Given precipitation distribution and simulated precipitation distribution; (**d**) Relative error between given and simulated result.

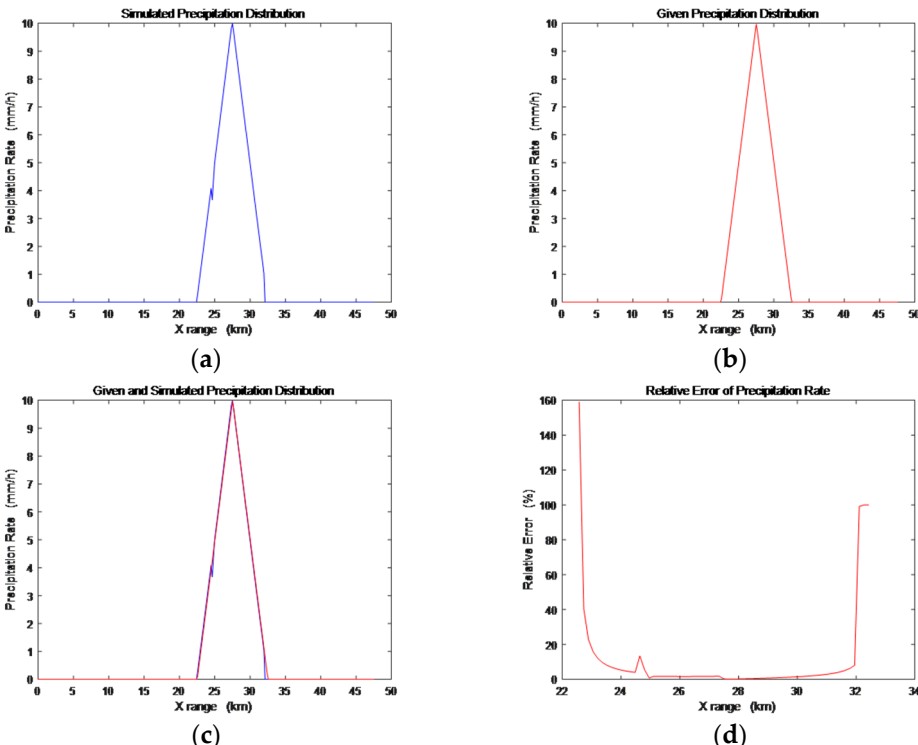

**Figure 4.** Triangular distribution diagrams for $R(x) = 10$ mm/h. (**a**) Given precipitation distribution; (**b**) Simulated precipitation distribution; (**c**) Given precipitation distribution and simulated precipitation distribution; (**d**) Relative error between given and simulated result.

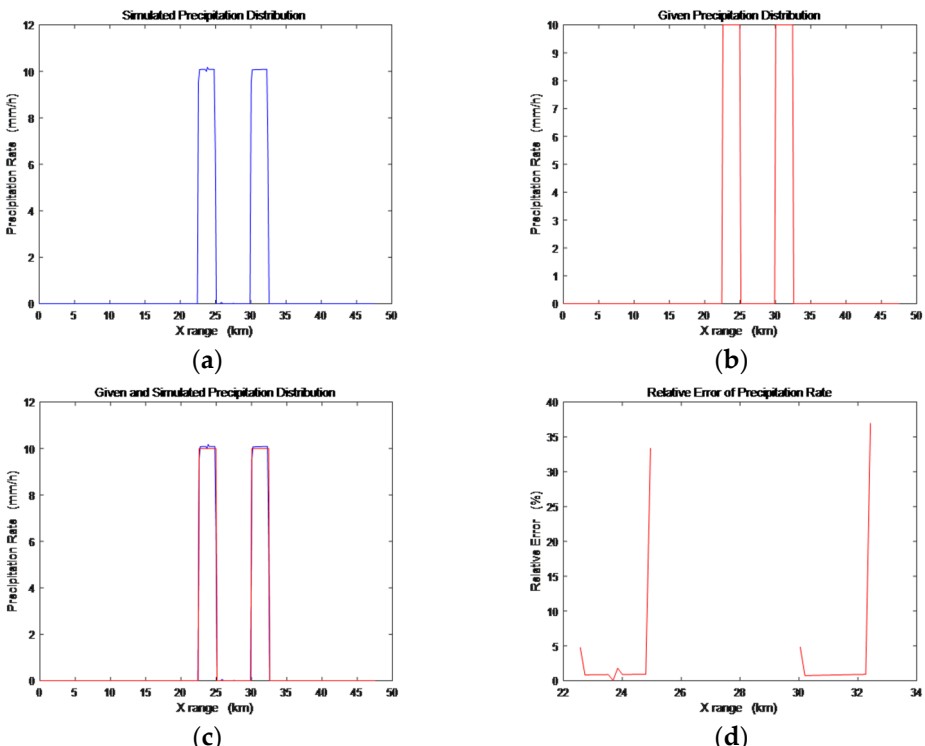

**Figure 5.** Rectangular bimodal distribution diagrams for $R(x) = 10$ mm/h. (**a**) Given precipitation distribution; (**b**) Simulated precipitation distribution; (**c**) Given precipitation distribution and simulated precipitation distribution; (**d**) Relative error between given and simulated results.

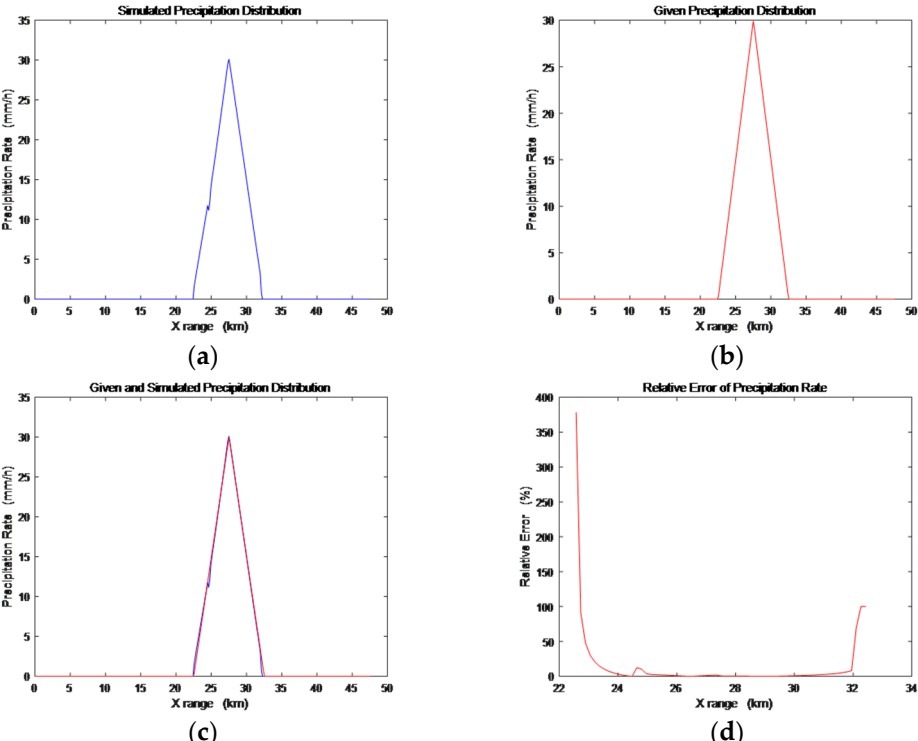

**Figure 6.** Triangular distribution diagrams for $R(x) = 30$ mm/h. (**a**) Given precipitation distribution; (**b**) Simulated precipitation distribution; (**c**) Given precipitation distribution and simulated precipitation distribution; (**d**) Relative error between given and simulated results.

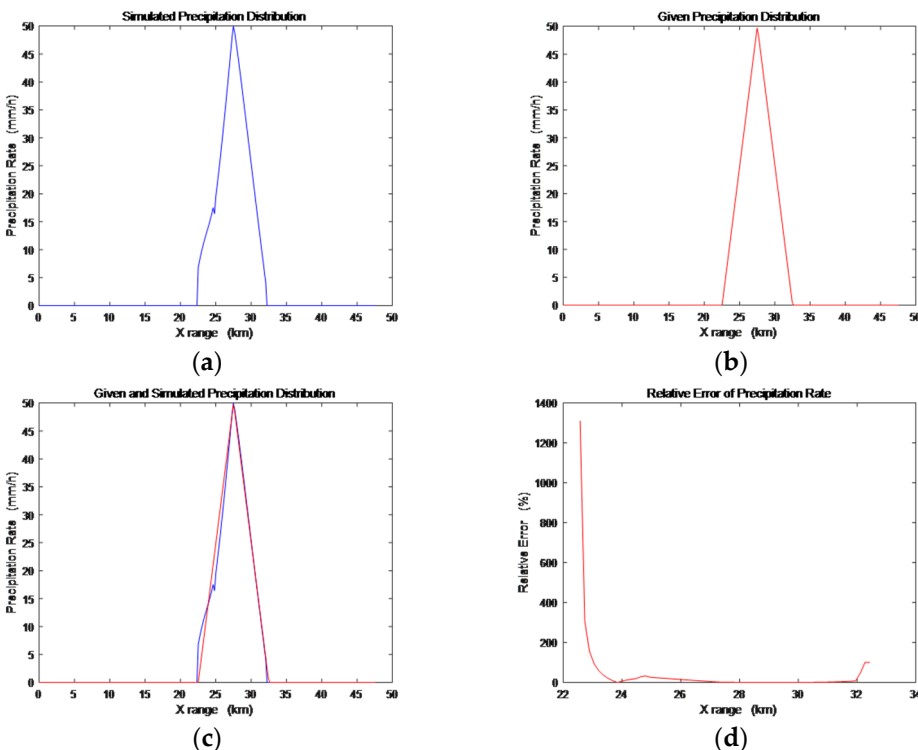

**Figure 7.** Triangular distribution diagrams for $R(x) = 50$ mm/h. (**a**) Given precipitation distribution; (**b**) Simulated precipitation distribution; (**c**) Given precipitation distribution and simulated precipitation distribution; (**d**) Relative error between given and simulated results.

**Table 2.** The attributes of the 6 examples.

|  | Maximum $R(x)$ (mm/h) | $d$ (km) | $w$ (km) | Shape of Rain-Cloud(s) |
|---|---|---|---|---|
| 1 | 10 | 0 | 10 | Rectangular |
| 2 | 10 | 3 | 10 | Trapezoidal |
| 3 | 10 | 5 | 10 | Triangular |
| 4 | 10 | 2.5 | 2.5 | Rectangular bimodal |
| 5 | 30 | 5 | 10 | Triangular |
| 6 | 50 | 5 | 10 | Triangular |

**Table 3.** Simulation result of the 6 examples.

|  | Maximum $\hat{R}(x)$ (mm/h) | $\tilde{R}(x)$ | $\hat{w}$ (km) | $\tilde{w}$ | Shape of Rain-Cloud(s) |
|---|---|---|---|---|---|
| 1 | 10.07 | 0.7% | 10.1731 | 1.731% | Rectangular |
| 2 | 10.08 | 0.8% | 9.8552 | 1.448% | Trapezoidal |
| 3 | 9.965 | 0.35% | 9.8552 | 1.448% | Triangular |
| 4 | 10.09 | 0.9% | 2.56 | 2.4% | Rectangular bimodal |
| 5 | 29.9 | 1% | 9.8552 | 1.448% | Triangular |
| 6 | 49.83 | 1.7% | 9.9255 | 0.745% | Triangular |

According to Figures 2–7, it is obvious that large errors occur at the edges of the precipitation area. The relative errors of the rest areas of the simulation results will be listed for each example.

**Example 1.** *Due to the non-linearity, fluctuations occur at each junction, which is $\hat{x}_R - n \cdot (z_t - z_0) \tan \theta$ or $\hat{x}_R - n \cdot z_t \tan \theta$, where n is a natural number. Except the right edge, $\hat{R}(x)$ is in the range of $[9.9, 10.07]$ mm/h in the precipitation area, and the maximum relative error is 1%.*

**Example 2.** *The error 8.073% occurs at the junction, $\hat{x}_R - z_t \tan\theta$, due to the non-linearity. In the rest of the precipitation area, the relative error of $\hat{R}(x)$ is less than 3%.*

**Example 3.** *The error at the junction, $\hat{x}_R - z_t \tan\theta$, is 13.47%. In the rest of the precipitation area, maximum relative error of $\hat{R}(x)$ is 1.7%.*

**Example 4.** *Due to the non-linearity, fluctuations occur at each junction, which is $\hat{x}_R - n \cdot (z_t - z_0) \tan\theta$ or $\hat{x}_R - n \cdot z_t \tan\theta$, where n is a natural number. Except the edges and the junctions, $\hat{R}(x)$ is in the range of $[10.07, 10.09]$ mm/h in the precipitation area, maximum relative error is 0.9%.*

**Example 5.** *The error at the junction, $\hat{x}_R - z_t \tan\theta$, is 12.41%. In the rest of the precipitation area, maximum relative error of $\hat{R}(x)$ is less than 2%.*

**Example 6.** *The error of $\hat{R}(x)$ around the junction $\hat{x}_R - z_t \tan\theta$ is 13.12%. In the rest of the precipitation area, maximum relative error of $\hat{R}(x)$ is 1.8%.*

## 5. Discussion

In Section 4, Examples 1–4 are the simulations under the maximum rain rate of 10 mm/h with different shapes of the rain-cloud cells, that is the rectangular rain-cloud cell, the trapezoidal rain-cloud cell, the triangular rain-cloud cell and the rectangular bimodal rain-cloud cell. It is obvious that the simulated result using this method is in good agreement with the given distribution. Besides different shapes of the rain-cloud cells, this method can also be applied to different rain rates. Examples 1, 5 and 6 are the simulations under the maximum rain rates of 10 mm/h, 30 mm/h and 50 mm/h with a triangular rain-cloud cell. The typical value 30 mm/h is chosen because the melting layer, the region where the precipitation changes from snow to rain, exists at lower rain rates, approximately below 30 mm/h [30]. Furthermore, the result has shown that the simulation result using this method is also in good agreement with the given one.

However, simulation errors exist. Large errors occur at the edges of the precipitation area, which are also where the biggest errors of the MOS method and the MOSVI method occur. For some demanding applications, more studies should be carried out to detect the edges of the precipitation area more accurately. Besides, fluctuations exist at the junctions of intervals introduced in Section 3 and Appendix B in detail. This is mainly because the unavoidable non-linearity of the function $p(x)$. Since the precipitation rate I(x) is derived from the function p(x), the simulated precipitation rate Î(x) is nonlinear as well. Thus, the fluctuations may occur at the junctions of the adjacent intervals. Furthermore, the fluctuations may affect the retrieval that follows.

It seems that the method introduced in this paper does not decrease the errors greatly, but taking a deep insight into the process of MOS method, great improvements can be sensed. The MOS method will determine the shape of the rain-cloud cell according to some of the features of the NRCS values. Furthermore, the final retrieval of the horizontal variation will be determined by using a certain equation, such as Equations (6) or (7) in this paper. This seems quite simple but the equation of the horizontal variation differs from shape to shape. To cover more shapes, large number of equations for the horizontal variation should be prepared in advance, which is a large workload at the early stages. Furthermore, for those rain-cloud cells which are not previously prepared, their shapes will be approximated to the closest one. It is clear that not all shapes can be covered previously, this method will add some errors when retrieving the uncovered shapes. The method introduced in this paper, however, can obtain the horizontal variation $I(x)$ within the range of interest analytically, which will increase the accuracy of the retrieval of the horizontal variation.

Another merit of this method is that it uses less empirical coefficients. Thus, this method can reduce the workload of the early stage and the simulation process. What is more, since each empirical coefficient is calculated under a certain rain rate, slant angle of the radar or other conditions, the fewer empirical coefficients the method uses, the more

scenarios in which the method can be easily applied. For the MOS and MOSVI method, the simulated widths of rectangular distributions, triangular distributions and trapezoidal distributions are calculated with empirical coefficients:

$$\hat{w} = 0.97(\hat{x}_{min} - \hat{x}_L), \tag{26}$$

$$\hat{w} = 1.61(\hat{x}_{min} - \hat{x}_L)^{0.93}, \tag{27}$$

$$\hat{w} = \left[ 0.97(\hat{x}_{min} - \hat{x}_L) + 1.61(\hat{x}_{min} - \hat{x}_L)^{0.93} \right]/2, \tag{28}$$

where $\hat{x}_{min}$ is the simulated result of the $x$-coordinate of the point whose NRCS reaches the minimum and $\hat{x}_L$ is the simulated result of $x$-coordinate of the left edge of the precipitation area. Furthermore, for the MOS method, the identification of the shape of the rain-cloud cell is also quite complex. First, the likelihood distance $d(c_H)$ is calculated with empirical coefficients:

$$d(c_H) = (x_{mSAR} - m_{cSAR})^T \cdot C_{cSAR}(x_{mPSAR} - m_{cPSAR}), \tag{29}$$

where $c_H$ is the shape class of the rain-cloud cell, $x_{mSAR}$ is the vector consisting of the measured observables, $m_{cSAR}$ is the mean vector and $C_{cSAR}$ is the covariance matrix. What makes it more complex is that in order to calculate the mean vector $m_{cPSAR}$ and the covariance matrix $C_{cPSAR}$, 11 simulated parameters for 50 possible NRCS responses of each $I(x)$ should be calculated, respectively. For four different shapes of the rain-cloud cells, as is used in this paper, at least 2200 parameters should be calculated, which greatly adds to the workload in terms of early preparation, time consumption and application limits of the retrieval process. It takes approximately 3 to 4 days using the statistical method, the MOS method, to retrieve the precipitation information on the whole [11] but less than 1 day using the proposed method to retrieve the precipitation information.

## 6. Conclusions

Through the rigorous formula derivation, this paper proposes an analytic method to obtain the horizontal variation suitable for the double-layer model with rain and snow layers. This method adds the accuracy of the retrieval when faced compared with MOS method. Furthermore, it depends less on empirical values, which means it can not only be applied to more scenarios, but it can also greatly save the time of preliminary preparation required by MOS method and MOSVI method, and improves the efficiency and accuracy of SAR precipitation inversion. At the same time, the errors of this method are acceptable.

It is clear that the method using SARs is more complex than that using precipitation radars (PRs), because PRs has high radial resolution and thus can provide three-dimensional precipitation information while SARs analyzes the scattering and attenuation along oblique path to obtain precipitation information. Therefore, like many other researchers, this paper carries on many simplifications, listed in Section 2.2, to make the method practical. Many considerations presented in this study can provide guidance for the development of such a retrieval algorithm. Furthermore, further research should focus on more realistic situations. The melting layer should be taken into consideration. Accurate information on the physical properties of the melting-snow layer is necessary to analyze the melting layer attenuation. The double-layer model used in this paper is obviously not representative of all precipitation structures, which requires further consideration of rainfall structures in different regions and situations. Thus, the method should be adjusted and applied to some practical situations to check its feasibility and accuracy further.

**Author Contributions:** Conceptualization, Y.X. and R.W.; methodology, T.L.; software, T.L. and X.Y.; validation, T.L. and X.Y.; formula derivation, T.L.; writing—original draft preparation, T.L.; writing—review and editing, Y.X., X.Y. and T.L.; funding acquisition, Y.X. All authors have read and agreed to the published version of the manuscript.

**Funding:** This research was funded by National Natural Science Foundation of China, grant number 62071286.

**Institutional Review Board Statement:** Not applicable.

**Informed Consent Statement:** Not applicable.

**Data Availability Statement:** Not applicable.

**Conflicts of Interest:** The authors declare no conflict of interest. The funders had no role in the design of the study; in the collection, analyses, or interpretation of data; in the writing of the manuscript, or in the decision to publish the results.

## Appendix A

This appendix is the analysis of radar backscattering echo path.

The radar backscattering echo path is shown in Figure A1. Let the x-coordinate of the point where the radar beam reaches the ground be x.

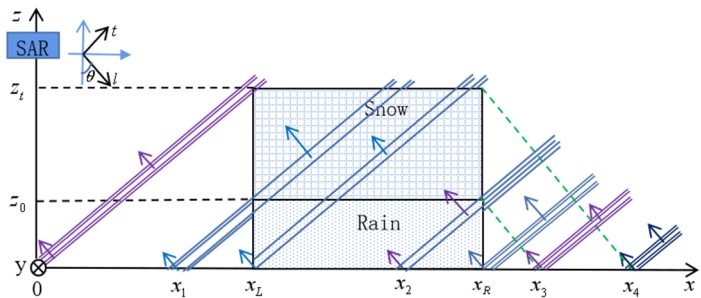

**Figure A1.** Radar echoes path diagram. The arrow direction is the backscattering echo direction. The microwave pulses emitted by the radar are considered as plane wave front slices, shown as pairs of lines.

When the microwaves emitted by the radar have not entered the precipitation area, that is, $x \in [-\infty, 0]$, there is surface backscattering echoes caused by the ground, without attenuation by the precipitation. The x-coordinates of the left and right limits of the attenuation path of scattering echo along l-direction and the left and right limits of the attenuation path of reflecting along t-direction caused by meteorological particles are:

$$x_{s_1} = 0, \tag{A1}$$

$$x_{s_2} = 0, \tag{A2}$$

$$x_{v_1} = 0, \tag{A3}$$

$$x_{v_2} = 0. \tag{A4}$$

The surface backscattering cross section and the volume backscattering cross section can be expressed as:

$$\sigma_{srf}(x) = \sigma_0, \tag{A5}$$

$$\sigma_{vol}(x) = 0. \tag{A6}$$

When the reflected microwaves begin to enter the snowfall area, that is, $x \in [0, x_1]$, the surface backscattering echoes caused by the ground are still unattenuated:

$$\sigma_{srf}(x) = \sigma_0. \tag{A7}$$

However, the volume backscattering echoes caused by snowfall appear:

$$x_{v_1} = x_L, \tag{A8}$$

$$x_{v_2} = x + z_t / \tan \theta, \tag{A9}$$

$$x_{v_3} = x_L, \tag{A10}$$

$$x_{v_4} = X. \tag{A11}$$

Since the volume backscattering echoes are only attenuated by the snowfall, the extinction coefficient is $k_s$ and the radar reflectivity is $\eta_s$. The volume backscattering cross section can be expressed as:

$$\sigma_{vol}(x) = \tan\theta \int_{x_{v_1}}^{x_{v_2}} exp\left(-2/\sin\theta \int_{x_{v_3}}^{x_{v_4}} k_s(\xi)d\xi\right)\eta_s(X)dX. \tag{A12}$$

For $x \in [x_1, x_L]$, the surface backscattering echoes caused by the ground are still unattenuated, there are snowfall backscattering echoes, and the rainfall backscattering echoes start to occur. The Equations (A8)–(A11) still hold, but the extinction coefficient k and the radar reflectivity $\eta$ differ:

$$k(\xi) = \begin{cases} k_r(\xi), & \xi \in [x_{v_3}, X - [z_0 - (X - x)\tan\theta]\tan\theta] \\ k_{s(\xi)}, & \xi \in [X - [z_0 - (X - x)\tan\theta]\tan\theta, x_{v_4}] \end{cases}, \tag{A13}$$

$$\eta(X) = \begin{cases} \eta_r(X), & X \in [x_{v_1}, x + z_0/\tan\theta] \\ \eta_s(X), & X \in [x + z_0/\tan\theta, x_{v_2}] \end{cases}. \tag{A14}$$

The volume backscattering cross section can be expressed as:

$$\sigma_{vol}(x) = \tan\theta \int_{x+z_0/\tan\theta}^{x_{v_2}} exp\left(-\frac{2}{\sin\theta}\int_{x_{v_3}}^{x_L} k_s(\xi)d\xi\right)\eta_s(X)dX +$$
$$\tan\theta \int_{x_{v_1}}^{x+z_0/\tan\theta} exp\left[-\frac{2}{\sin\theta}\left(\int_{x_{v_3}}^{X-[z_0-(X-x)\tan\theta]\tan\theta} k_s(\xi)d\xi + \int_{X-[z_0-(X-x)\tan\theta]\tan\theta}^{x_{v_4}} k_r(\xi)d\xi\right)\right]\eta_r(X)dX. \tag{A15}$$

When the microwaves emitted by the radar begin to enter the snowfall area, that is, $x \in [x_L, x_R]$, the surface backscattering echoes caused by the ground begin to be attenuated, and there are precipitation backscattering echoes:

$$x_{s_1} = max\{x_L, x - z_t \tan\theta\}, \tag{A16}$$

$$x_{s_2} = x. \tag{A17}$$

$$x_{v_1} = x, \tag{A18}$$

$$x_{v_2} = min\{x_R, x + z_t/\tan\theta\}. \tag{A19}$$

$$x_{v_3} = max\{x_L, X - [z_t - (X - x)\tan\theta]\tan\theta\}, \tag{A20}$$

$$x_{v_4} = X. \tag{A21}$$

The surface backscattering cross section can be expressed as:

$$\sigma_{srf}(x) = \sigma_0 \cdot exp\left[-2/\sin\theta\left(\int_{x_{s_1}}^{x-z_0\tan\theta} k_s(\delta)d\delta + \int_{x-z_0\tan\theta}^{x_{s_2}} k_r(\delta)d\delta\right)\right]. \tag{A22}$$

The volume backscattering cross section can be expressed by Equation (A15).

For $x \in [x_R, x_3]$, the surface backscattering echoes caused by the ground are attenuated but there are no precipitation backscattering echoes:

$$x_{s_1} = x - z_t \tan\theta, \tag{A23}$$

$$x_{s_2} = x_R, \tag{A24}$$

$$x_{v_1} = 0, \tag{A25}$$

$$x_{v_2} = 0. \tag{A26}$$

The surface backscattering cross section can be expressed by (A22) and the volume backscattering cross section can be expressed as:

$$\sigma_{vol}(x) = 0. \tag{A27}$$

For $x \in [x_3, x_4]$, the surface backscattering echoes caused by the ground are only attenuated by snowfall and there are no volume backscattering echoes:

$$x_{s_1} = x - z_t \tan \theta, \tag{A28}$$

$$x_{s_2} = x_R. \tag{A29}$$

$$x_{v_1} = 0, \tag{A30}$$

$$x_{v_2} = 0. \tag{A31}$$

The surface backscattering cross section can be expressed as:

$$\sigma_{srf}(x) = \sigma_0 \cdot exp\left(-2/\sin\theta \int_{x_{s_1}}^{x_{s_2}} k_s(\delta)d\delta\right), \tag{A32}$$

The volume backscattering cross section can be expressed as:

$$\sigma_{vol}(x) = 0. \tag{A33}$$

When the microwaves emitted by the radar finally leave the precipitation area, that is, $x \in [x_4, \infty]$, there is only unattenuated surface backscattering echoes caused by the ground.

$$x_{s_1} = 0, \tag{A34}$$

$$x_{s_2} = 0, \tag{A35}$$

$$x_{v_1} = 0, \tag{A36}$$

$$x_{v_2} = 0. \tag{A37}$$

The surface backscattering cross section and the volume backscattering cross section can be, respectively, expressed as:

$$\sigma_{srf}(x) = \sigma_0, \tag{A38}$$

$$\sigma_{vol}(x) = 0. \tag{A39}$$

**Appendix B**

This appendix is the mathematical details on VIE analytic formulations.

Using the function $p(x)$ defined in Section 3, the extinction coefficients of rain, the radar reflectivity of rain and snow, together with the precipitation rate can also be expressed as:

$$k_r(x) = \frac{\sin\theta}{2}\frac{p_x'(x)}{p(x)}, \tag{A40}$$

$$\eta_r(x) = a_{1r}\left[\frac{\sin\theta}{2a_r}\frac{p_x'(x)}{p(x)}\right]^{b_{1r}}, \tag{A41}$$

$$\eta_s(x) = a_{1s}\left[\frac{\sin\theta}{2a_r}\frac{p_x'(x)}{p(x)}\right]^{b_{1s}}, \tag{A42}$$

$$I(x) = \frac{\sin\theta}{2a_r}\frac{p_x'(x)}{p(x)}, \tag{A43}$$

where $p_x'(x)$ is the derivative of the function $p(x)$.

Since the linear model of the extinction coefficient is adopted, we can obtain:

$$k_s(x) = \frac{a_s}{a_r} k_r(x). \tag{A44}$$

Substitute (A40)–(A42) and (A44) into the Formula (18), we can obtain:

$$\sigma_{SAR}(x) = \sigma_0 \left[ \frac{p(x-z_t \tan\theta)}{p(x-z_0 \tan\theta)} \right]^{\frac{a_s}{a_r}} \cdot \frac{p[x-z_0 \tan\theta]}{p(x)} + G_1 \int_{x+z_0/\tan\theta}^{x+z_t/\tan\theta} \left[ \frac{p\{X-[z_t-(X-x)\tan\theta]\tan\theta\}}{p(X)} \right]^{\frac{a_s}{a_r}} \cdot \left[ \frac{p_X'(X)}{p(X)} \right]^{b_{1s}} dX +$$
$$G_2 \int_x^{x+z_0/\tan\theta} \left[ \frac{p\{X-[z_t-(X-x)\tan\theta]\tan\theta\}}{p\{X-[z_0-(X-x)\tan\theta]\tan\theta\}} \right]^{\frac{a_s}{a_r}} \cdot \frac{p\{X-[z_0-(X-x)\tan\theta]\tan\theta\}}{p(X)} \cdot \left[ \frac{p_X'(X)}{p(X)} \right]^{b_{1r}} dX. \tag{A45}$$

where $p_X'(X)$ is the derivative of function $p(X)$ and

$$G_1 = a_{1s} \tan\theta \left( \frac{\sin\theta}{2a_r} \right)^{b_{1s}}, \tag{A46}$$

$$G_2 = a_{1r} \tan\theta \left( \frac{\sin\theta}{2a_r} \right)^{b_{1r}}. \tag{A47}$$

To avoid the arbitrariness of the choice of $x$, we shift $x$ to $x + z_t \tan\theta$:

$$\sigma_{SAR}(x + z_t \tan\theta) = \sigma_0 \left[ \frac{p(x)}{p[x+(z_t-z_0)\tan\theta]} \right]^{\frac{a_s}{a_r}} \cdot \frac{p[x+(z_t-z_0)\tan\theta]}{p(x+z_t \tan\theta)} +$$
$$G_1 \int_{x+z_0/\tan\theta+z_t \tan\theta}^{x+z_t/\tan\theta+z_t \tan\theta} \left[ \frac{p\{X-[z_t-(X-x-z_t \tan\theta)\tan\theta]\tan\theta\}}{p(X)} \right]^{\frac{a_s}{a_r}} \cdot \left[ \frac{p_X'(X)}{p(X)} \right]^{b_{1s}} dX +$$
$$G_2 \int_{x+z_t \tan\theta}^{x+z_0/\tan\theta+z_t \tan\theta} \left[ \frac{p\{X-[z_t-(X-x-z_t \tan\theta)\tan\theta]\tan\theta\}}{p\{X-[z_0-(X-x-z_t \tan\theta)\tan\theta]\tan\theta\}} \right]^{\frac{a_s}{a_r}} \cdot \frac{p\{X-[z_0-(X-x-z_t \tan\theta)\tan\theta]\tan\theta\}}{p(X)} \cdot \left[ \frac{p_X'(X)}{p(X)} \right]^{b_{1r}} dX. \tag{A48}$$

Introduce a change of the integration variable, $t$, in Equation (A48):

$$t = X - [z_t - (X - x - z_t \tan\theta)\tan\theta]\tan\theta, \tag{A49}$$

$$\mathrm{d}X = \cos^2\theta \mathrm{d}t, \tag{A50}$$

and then we can obtain:

$$\sigma_{SAR}(x + z_t \tan\theta) = \sigma_0 \left[ \frac{p(x)}{p[x+(z_t-z_0)\tan\theta]} \right]^{\frac{a_s}{a_r}} \cdot \frac{p[x+(z_t-z_0)\tan\theta]}{p(x+z_t \tan\theta)} +$$
$$G_1 \cos^2\theta \int_{x+z_0/\tan\theta+z_0 \tan\theta}^{x+z_t/\tan\theta+z_t \tan\theta} p(t)^{\frac{a_s}{a_r}} K_1[X(x,t)]dt + G_2 \cos^2\theta \int_x^{x+z_0/\tan\theta+z_0 \tan\theta} p(t)^{\frac{a_s}{a_r}} \cdot K_2[X(x,t)]dt, \tag{A51}$$

where

$$K_1(X) = \left[ \frac{1}{p(X)} \right]^{\frac{a_s}{a_r}} \cdot \left[ \frac{p_X'(X)}{p(X)} \right]^{b_{1s}}, \tag{A52}$$

$$K_2(X) = \left[ \frac{1}{p[t+(z_t-z_0)\tan\theta]} \right]^{\frac{a_s}{a_r}} \cdot \frac{p(t+(z_t-z_0)\tan\theta)}{p(X)} \cdot \left[ \frac{p_X'(X)}{p(X)} \right]^{b_{1r}}, \tag{A53}$$

$$X(x,t) = t\cos^2\theta + x\sin^2\theta + z_t \tan\theta. \tag{A54}$$

Then, we transpose Equation (A51) to solve $p(x)$:

$$p(x)^{\frac{a_s}{a_r}} = f_1(x) - \lambda_1 \int_{x+z_0/\tan\theta+z_0 \tan\theta}^{x+z_t/\tan\theta+z_t \tan\theta} K_3(X)p(t)^{\frac{a_s}{a_r}} dt - \lambda_2 \int_x^{x+z_0/\tan\theta+z_0 \tan\theta} K_4(X)p(t)^{\frac{a_s}{a_r}} dt, \tag{A55}$$

where

$$f_1(x) = \frac{\sigma_{SAR}(x+z_t \tan\theta) \cdot p(x+z_t \tan\theta) \cdot p[x+(z_t-z_0)\tan\theta]^{\frac{a_s}{a_r}-1}}{\sigma_0}, \tag{A56}$$

$$K_3(X) = K_1(X) \cdot p[x + (z_t - z_0) \tan \theta]^{\frac{a_s}{a_r} - 1} \cdot p(x + z_t \tan \theta), \tag{A57}$$

$$K_4(X) = K_2(X) \cdot p[x + (z_t - z_0) \tan \theta]^{\frac{a_s}{a_r} - 1} \cdot p(x + z_t \tan \theta), \tag{A58}$$

$$\lambda_1 = \frac{G_1 \cos^2 \theta}{\sigma_0}, \tag{A59}$$

$$\lambda_2 = \frac{G_2 \cos^2 \theta}{\sigma_0}. \tag{A60}$$

The iteration processes are explicitly shown as follows.

For $x \geq x_m$, we have

$$p(x) = 1. \tag{A61}$$

For $x \geq x_m - (z_t - z_0) \tan \theta$, we have

$$I(x + (z_t - z_0) \tan \theta) = 0, p(x + (z_t - z_0) \tan \theta) = 1. \tag{A62}$$

We can obtain:

$$p(x) = \left[ \frac{\sigma_{SAR}(x + z_t \tan \theta)}{\sigma_0} \right]^{\frac{a_r}{a_s}}. \tag{A63}$$

For $x \geq x_m - z_t \tan \theta$, we have $I(x + z_t \tan \theta) = 0$, $p(x + z_t \tan \theta) = 1$. Assume that $z_t \geq 2z_0$, then we have:

$$p[x + (z_t - z_0) \tan \theta] = \left[ \frac{\sigma_{SAR}(x + 2z_t \tan \theta - z_0 \tan \theta)}{\sigma_0} \right]^{\frac{a_r}{a_s}}. \tag{A64}$$

We can obtain:

$$p(x) = \left\{ \frac{\sigma_{SAR}(x + z_t \tan \theta)}{\sigma_0} \left[ \frac{\sigma_{SAR}(x + 2z_t \tan \theta - z_0 \tan \theta)}{\sigma_0} \right]^{1 - \frac{a_r}{a_s}} \right\}^{\frac{a_r}{a_s}}. \tag{A65}$$

Continuing such iteration, $p(x)$ within the range of interest can be obtained. Furthermore, through Equation (A43), we can then determine $I(x)$ within range of interest.

According to the geometric characteristics of rectangles, triangles and trapezoids, once we know the precipitation rate, we can obtain the precipitation distribution. A rough method to distinguish the horizontal variation can be used here [11]. We first calculate two parameters:

$$dI_1(x) = \frac{I(x + \Delta x) - I(x)}{\Delta x}, \tag{A66}$$

$$dI_2(x) = \frac{I(x + 2 \cdot \Delta x) - I(x + \Delta x)}{\Delta x}. \tag{A67}$$

where $\Delta x$ is a small increment of $x$, $dI_1(x)$ and $dI_2(x)$ are the slopes at two successive points, respectively.

When the slope is considerably large, taking the typical value 30 as an example:

$$dI_1(x) > 30, \tag{A68}$$

the horizontal variation can be seen as rectangular. The $x$-coordinate of the left point of the precipitation area $\hat{x}_L = x$. Similarly, the $x$-coordinate of the right point of the precipitation area $\hat{x}_R$ is the point at which $dI_1(x)$ is considerably small. The width of the precipitation area $\hat{w}$ can be consequently calculated through:

$$\hat{w} = \hat{x}_R - \hat{x}_L. \tag{A69}$$

When

$$dI_1(x) \geq 2 \cdot dI_2(x), \tag{A70}$$

$$dI_2(x) \geq 0, \tag{A71}$$

the horizontal variation can be seen as trapezoidal.
When

$$dI_1(x) \geq 2 \cdot dI_2(x), \tag{A72}$$

$$dI_2(x) \leq 0, \tag{A73}$$

the horizontal variation can be seen as triangular. For trapezoidal rain-cloud cells and triangular rain-cloud cells, the $x$-coordinate of the left edge of the precipitation area $\hat{x}_L$, the $x$-coordinate of the right edge of the precipitation area $\hat{x}_R$ and the width of the precipitation area $\hat{w}$ can be calculated similar to those of the rectangular rain-cloud cells.

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
