# Peer review of "An Analytic Solution to Precipitation Attenuation Expression with Spaceborne Synthetic Aperture Radar Based on Volterra Integral Equation"

_remotesensing, doi:10.3390/rs14020357_

Round 1

Reviewer 1 Report

The manuscript deals with an analytic solution to precipitation attenuation expression with spaceborne SAR based on Volterra integral equation. Authors tried to reduce the time for calculation and simplify related equations. I do not think that the results obtained from this research is clearly described in the current manuscript and the equations are too many explained in the main Section. Therefore, I would like to recommend that it should be modified before publication.

Comments

  1. Lines 83-85: This sentence would move to another Section like Conclusions or Abstract
  2. Lines 187-191: In the Table 1, I understand that power-law relation shows better performance than that of linear model. Authors mentioned that they selected linear model because of its less complexity. I think that authors need to describe more clear reason why they choose the linear model.
  3. Section 2.5 and Section 3: I think that these two sections had better move to Appendix.
  4. Results: Authors used 6 scenarios for the analysis. I would like to know if these 6 scenarios with different R, w, d are representative for real precipitation. And authors had better make Table and the main results should be placed into the Table for better understanding.
  5. Conclusions: I think that authors need to describe the conclusions obtained from the results more detail and quantitatively. I can not understand clearly what the main results are. For example, if we use the proposed method, how much time we could save?

Author Response

We are really grateful to the reviewer for all the constructive remarks and useful suggestions, which has significantly raised the quality of the manuscript. Each suggested revision and comment was accurately incorporated and considered. Below the comments of the reviewers are the responses point by point. Please see the attachment.

Reviewer 2 Report

The manuscript is well organized, the results are quite interesting and it can be accepted in the present form.

Author Response

We are really grateful to your careful review.

Reviewer 3 Report

Review of "An analytic solution to precipitation attenuation expression with Spaceborne Synthetic Aperture Radar based on Volterra integral equation"

by  Ting Luo, Yanan Xie, Rui Wang , and Xueying Yu

    The authors proposed an improved analytic method for the horizontal distribution of precipitation. Their method seems to be very helpful as a starting point for simulating observing the real distribution of precipitation. As they pointed out, however, many assumptions are still adopted to derive the equation. I hope they develop the solution for the possible practical formula for the actual situation of precipitation.

    In the real world, precipitation's spatial distribution is determined in association with the type of meteorological phenomena such as stratiform and convective styles of clouds accompanying rain and snow.

    Their figure 5, 7, and 8 show the different amounts of given and simulated errors. It might be much better to show the percentage ratio of errors to the given amount or intensity of the precipitation. In meteorology, the error in magnitude for precipitation observation is usually depicted as a fractional quantity compared with the observed precipitation values.

    And also, they must consider that the two-dimensional technique is how accurately the three-dimensional distribution in the real world.

    I like to say that the manuscript can be published with a minor correction for the main text.

Author Response

(The authors gave the same response as above.)

Round 2

Reviewer 1 Report

I think that the revised manuscript was modified according to my comments. One minor thing is that Table 3 and Table 4 could be one table for the simplication and move to the last paragraph of the Section 4 (line 276).

Author Response

Dear Reviewer,

We are really grateful to you for the constructive remarks, which has remarkably raised the quality of the manuscript. Each suggested comment was carefully considered and the revision is made accordingly. Below the comment is the response. Please see the attachment.

Yours sincerely,

Ms. Luo
